# Effects of the Family Nurse Partnership on all eligible mothers: a data linkage cohort study in England

**Katie Harron**[1]*, **Francesca Cavallaro**[2], **Jan van der Meulen**[3], **Eilis Kennedy**[4], **Ruth Gilbert**[1]

1 UCL Great Ormond Street Institute of Child Health, London, United Kingdom, 2 The Health Foundation, London, United Kingdom, 3 London School of Hygiene & Tropical Medicine, London, United Kingdom, 4 The Tavistock and Portman NHS Foundation Trust, London, United Kingdom

* k.harron@ucl.ac.uk

## Abstract

### Background

An intensive programme of home visiting, the Family Nurse Partnership (FNP), is received by around one in four first-time adolescent mothers in selected areas in England. During home visits, nurses support mothers to make choices about healthy pregnancies, improving child development, and fulfilling their own aspirations and ambitions. Evidence is needed of the wider effects of the FNP, including for mothers not enrolled in the programme (who might experience unintended effects). We evaluated child and maternal outcomes for all eligible mothers giving birth before, during, and after the period in which FNP was active in local areas.

### Methods

We created a linked cohort of 237,185 eligible mothers, aged 13-19, who gave birth between April 2010 and March 2019, and who had a first antenatal booking appointment (or a date of 28 completed weeks of gestation, if missing) when FNP was active in their area. We used administrative hospital data to identify unplanned maternal/child hospitalisations up to 2 years after birth for children born and mothers delivering before, during and after FNP was active. Generalised linear models were used to adjust for background regional time trends, maternal characteristics, and clustering of outcomes within residential areas.

### Results

We found no evidence of differences in unplanned hospital admissions for children born during the FNP period (36.9% versus 36.0%, relative risk [RR] 1.01; 95% CI 0.99-1.02), or after FNP was active (37.1%, RR 1.0; 95% CI 0.95-1.06), compared with those born before FNP was active. There was no evidence of differences in child admissions for maltreatment/injury-related diagnoses or for maternal admissions for adversity-related diagnoses.

**Data availability statement:** Individual data from this study cannot be shared due to data sharing agreements with NHS England and the Department for Education. HES and FNP data can be requested through NHS Digital and NPD can be requested through the Department for Education (https://digital.nhs.uk/services/data-access-request-service-dars and https://www.find-npd-data.education.gov.uk). This data can only be assessed on secure servers, with specific project and researcher permissions.

**Funding:** This study was supported by funding from NIHR Health Services Delivery & Research (17/99/19), and the authors were in part supported by the NIHR through the Great Ormond Street Hospital Biomedical Research Centre and the NIHR Children and Families Policy Research Unit and Senior Investigator award for RG. The funders had no role in study design, data collection and analysis, decision to publish, or preparation of the manuscript.

**Competing interests:** The authors have declared that no competing interests exist.

## Conclusion

Child and maternal outcomes were similar before, during and after FNP active periods, suggesting that the FNP did not have a wider impact on outcomes in all eligible mothers, including those not participating in the FNP.

## Introduction

Despite reductions in adolescent pregnancies over recent decades, there remains a significant population of young and vulnerable mothers in England who are at risk of adverse birth, child and maternal outcomes.[1,2] There is some evidence that early targeted interventions can help improve outcomes including the prevention of child maltreatment and injuries.[3] The most promising intervention in England is the Family Nurse Partnership, which aims to improve birth outcomes, child health and development, and to promote their the mothers' aspirations and ambitions and their ability to be economically self-sufficient.[4]

Evidence for the effectiveness of the FNP stems from three randomised controlled trials, carried out in the United States, which showed improvement in all primary child health and development outcomes and in some of the maternal outcomes.[5–8] However, a randomised controlled trial of the FNP in England (Building Blocks) found no evidence of effect on the four primary outcomes, including smoking in late pregnancy, birthweight, second pregnancy within 24 months of first birth, and rates of visits to the accident and emergency (A&E) department or unplanned hospital admissions within 24 months of birth.[9] Improvements were observed for some secondary and medium-term outcomes, including maternally reported child cognitive and language development and improved school readiness.[10,11] Despite the lack of research evidence supporting its effectiveness, there remains strong local support for the FNP programme.[4,12,13]

There is a need to understand the whole area effects of commissioning FNP, considering outcomes for all eligible women (not only those who participated in the programme), for several reasons.[14] Firstly, in areas in which the FNP programme is available, it is offered to only around one in four eligible mothers.[15] Enrolled mothers tend to be more vulnerable than those not enrolled (younger; more likely to have been in care or have a child protection plan, be recorded as having Special Educational Needs provision, live in the most deprived quintile according to Income Deprivation Affecting Children Index, have been excluded or be persistently absent at school).[16] In some areas, efforts have been made to tailor the programme more closely to the needs of individual mothers and their families, and to extend eligibility to older mothers with other indicators of risk.[17] We do not yet know if the FNP programme diverts resources away from the usual care that an adolescent mother would receive or if there are any other unintended consequences of the programme (Table 1).[18] The FNP programme is licensed, and involves protection of licensed materials. Qualitative research suggests that professionals are concerned about the licensed nature of FNP and not being able to share freely with wider health visiting colleagues or to use it to change practice across services.[19] Despite this, there may still be benefits to other services, for example if trained family nurses take on other roles and disseminate trauma-informed approaches, either while the FNP is being delivered, or after it has stopped being delivered in a local area. Indeed, evidence from the Building Blocks trial demonstrated that health visitors in the control arm conducted many more visits (~16) than would typically have been delivered in usual care prior to the trial.[12]

Secondly, whilst research using linked administrative data on the whole population of mothers participating in the FNP has the potential to provide evidence of the effect of the

**Table 1. Examples of potential benefits and disadvantages of the Family Nurse Partnership to the wider area, effecting both eligible but non-participating mothers and ineligible mothers.**

| | Potential benefits | Potential disadvantages |
|---|---|---|
| Staffing | Attracts more expert nurses to frontline community work | Drains capacity from business-as-usual health visiting |
| Knowledge of practice | Sharing of learning, e.g., on trauma-informed approaches | Confusion about/ restriction of the most appropriate tools to use for a specific population |
| Connection/displacement of related services | Shared knowledge of relevant services, e.g., midwifery, social care, mental health, early day care | Eligible mothers might be assumed to be receiving support from the FNP and therefore may be less likely to be referred to additional services; potential reductions in funding to other services due to the expense of commissioning FNP |

programme on a much larger scale than possible within a randomised trial, such observational studies face a number of challenges.[20] The main challenge is to overcome the confounding by indication that is inherent in study designs comparing mothers who are and are not enrolled in the programme (since those who are enrolled are a highly selected population). [15,16] The lack of effect observed in an observational study using linked administrative data for all FNP participants between 2010-2019 in England may be a consequence of such residual confounding.[16]

Evaluating the effect of the FNP on all eligible mothers would overcome the issues of residual confounding in earlier studies, and would also provide an insight into the overall effect of the programme, balancing potential positive spillover effects and unintended negative effects on those who are not enrolled.[21] We therefore aimed to estimate the wider effects of the FNP by evaluating child and maternal healthcare outcomes for all eligible mothers, including those who were not enrolled in the FNP, to determine whether the whole population of adolescent mothers were overall better or worse off during periods in which the FNP was active in each area, compared to before and after. This research will contribute to a broader evidence base to help inform decisions about offering more intensive, targeted programs to young and vulnerable mothers versus broader reaching universal services.[21]

## Methods

### Ethical approvals

Support for this study was obtained from Nottingham Research Ethics Committee (ref 18/ EM/0014). Data used for this study was pseudonymised and there was no requirement for consent.

### Data sources and linkage

We used linked hospital records from Hospital Episode Statistics (HES), education and social care records from the National Pupil Database (NPD), and FNP programme data (from the FNP Information System) for mothers and their children. HES is an administrative dataset that contains details of all hospital admissions (from 1997), outpatient appointments (from 2003) A&E department visits (from 2010) at NHS hospitals in England.[22] NPD includes information on pupils attending state schools and their contact with social care services in England.

HES records were used to identify the study population of mothers and their children, and were linked to the NPD by the Department for Education using a matching algorithm requiring agreement (full or "fuzzy") on names, date of birth and postcode (84.1% of mothers in HES were linked to an NPD record).[23–25] Linked NPD data provided information on additional maternal characteristics not captured in HES. The linkage is described in more

detail elsewhere.[26] We obtained information about FNP participation dates from the FNP Information System (98.5% of FNP mothers were linked to a HES record).

## Study population

Our study population included all first-time mothers aged between 13 and 19 years at their last menstrual period who had a live birth in England between 1 April 2010 and 31 March 2019. 136/152 Local Authorities, a level of local government responsible for services including education, social services and public health in England, had an active FNP site at some point between 2010 and 2019 (Table S1).[27] Our study population included all mothers eligible for the FNP, including those who were enrolled in the FNP, and those were not offered support from the FNP programme as there were not enough places for all eligible mothers, or because they were eligible before or after the FNP enrolment period.

## Exposure

The FNP enrolment period for each area defined by Local Authority boundaries was determined from the FNP Information System data which includes information on the first and last dates of enrolment within each area. Enrolment periods were additionally checked with the FNP National Unit (who are responsible for delivering strategic oversight and support for the quality delivery and development of the FNP programme in England). The periods in which FNP was active varied geographically (Table S1).

For each mother, we determined eligibility for the programme based on whether the pregnancy occurred in a period before or during the FNP was active, or after the FNP had stopped being offered, in the Local Authority in which she lived. Exposure was therefore determined by the date of first antenatal booking appointment (or a date of 28 completed weeks of gestation, if missing) and whether or not the FNP was active in their area at this date.

## Outcomes

FNP visits continue to be offered up to two years after birth, and so we were primarily interested in outcomes during this period. Previous analysis of the FNP data has shown that rates of any unplanned admissions and unplanned admissions for injury or maltreatment-related diagnoses in the child and unplanned admissions for adversity in the mother were slightly increased in mothers participating in the FNP, but that FNP mothers were less likely to have a subsequent delivery within 18 months of the index birth than those not enrolled.[16]

We therefore evaluated unplanned hospital admissions for children and mothers up to two years after birth, and specifically those with codes indicating injury or maltreatment-related diagnoses for the child, or adversity-related diagnosis (including substance misuse, self-harm or violence) for the mother. We created binary variables according to whether each child or mother had one or more admissions captured in HES using published code lists (Tables S2–S3). We also examined the effect of an FNP programme being active in the area on subsequent pregnancies within 18 months of first live birth.

To help to identify whether any differences were due to time-varying confounding, we also included low birth weight (<2500g) as a further outcome, as based on previous evidence, we did not expect that the FNP programme would have any effect on birth weight for mothers who were enrolled or those who were not.[28] Any differences in rates of low birth weight seen across periods in which FNP was active or not could therefore indicate that there were reasons other than the FNP programme being in place that might explain differences in the primary outcomes.

## Covariates

We used information from hospital admissions and A&E department attendances for the mother up to 2 years before 20 weeks of gestation (i.e., the date by which the majority of mothers have their antenatal booking appointment). For mothers whose HES record had been linked to NPD, data on maternal school attainment at Key Stage 2 (age 11) and Key Stage 4 (age 16) were included, alongside information on Free School Meals, Special Educational Needs provision, and absences and exclusions.[29] Covariates are listed in full in Table S4.

## Statistical analysis

We first described the maternal characteristics of eligible mothers whose antenatal booking appointment (or 28th completed week of pregnancy) was within the periods before, during, and after the FNP was active in their area.

We applied an interrupted time-series design using generalised linear models to estimate risk ratios to compare outcomes for mothers before, during and after the FNP was active in each area defined by Local Authority boundaries. We represented time trends by including calendar month in the model, centred around the date at which FNP became active in each area. An indicator variable was used to represent whether the antenatal booking appointment (or 28th week of pregnancy, if missing) was within the period before, during, or after FNP was active in the area. We also included an interaction between calendar month and the indicator representing an active FNP programme, to test for differences in time trends during the three periods.

Models were additionally adjusted for maternal characteristics prior to 20 weeks of gestation (Table S4) and quarter of year of delivery (to allow for seasonality). Multi-level models were used to account for clustering of outcomes within each Local Authority and included random intercepts to allow baseline risks to vary across Local Authorities; the effect of an active FNP programme being in place was assumed to be contract across all areas.

Data for this study were last accessed on 7/1/24. Authors did not have access to any information that could identify individual participants during or after data collection.

## Results

Of the 203,045 eligible mothers giving birth between April 2010 and March 2019 (S1 Fig), 57,480 (28%) were eligible before the FNP was active in their Local Authority, 130,485 (64%) were eligible during the period in which FNP was active, and 15,080 (7%) were eligible after the FNP had stopped. The characteristics of these different groups of mothers varied, as they were eligible in different calendar periods and in different areas (Table 2).

Compared with mothers eligible before FNP was active, there was no evidence of a difference in unplanned hospital admissions up to age two for children born to mothers eligible during an active FNP period (36.9% versus 36.0%, adjusted relative risk [aRR] 1.01; 95% CI 0.99-1.02) or after FNP had stopped (37.1%, aRR 1.0; 95% CI 0.95-1.06) (Table 3). There was no evidence of a difference in admissions for maltreatment/injury-related diagnoses up to age two or maternal admissions for adversity-related diagnosis across periods (Table 2). Differences in the rate low birth weight were not meaningful across periods (aRR 1.00; 95% CI 0.99-1.00 comparing births during versus before FNP was active and 1.01; 95% CI 1.01-1.02 comparing births after versus before FNP was active).

Interactions between calendar month and FNP period were not significant in any models, indicating that trends in outcomes were not affected by whether the FNP was active or not in a particular area (Figs 1–3).

**Table 2. Maternal characteristics of the study population eligible before, during or after the FNP enrolment period.**

| | Mothers before enrolment period | | Mothers during enrolment period | | Mothers after enrolment period | |
|---|---|---|---|---|---|---|
| | N | % | N | % | N | % |
| **Total** | 57480 | 100 | 130485 | 100 | 15080 | 100 |
| **Maternal age at birth (years)[1]** | | | | | | |
| 13-15 | 1230 | 2.1 | 2685 | 2.1 | 300 | 2.0 |
| 16-17 | 11945 | 20.8 | 26075 | 20.0 | 2830 | 18.8 |
| 18-19 | 32050 | 55.8 | 72500 | 55.6 | 8790 | 58.3 |
| 20 | 12255 | 21.3 | 29225 | 22.4 | 3160 | 21.0 |
| **Ethnicity** | | | | | | |
| White | 50380 | 87.6 | 109880 | 84.2 | 12405 | 82.3 |
| South Asian | 1515 | 2.6 | 3700 | 2.8 | 480 | 3.2 |
| Black | 1825 | 3.2 | 4655 | 3.6 | 345 | 2.3 |
| Mixed/other | 2385 | 4.1 | 6845 | 5.2 | 995 | 6.6 |
| Unknown | 1370 | 2.4 | 5415 | 4.1 | 865 | 5.7 |
| **Area-level deprivation (quintile of IMD)** | | | | | | |
| Least deprived | 4305 | 7.5 | 6810 | 5.2 | 695 | 4.6 |
| 2 | 5970 | 10.4 | 10405 | 8.0 | 1110 | 7.4 |
| 3 | 9085 | 15.8 | 17860 | 13.7 | 1870 | 12.4 |
| 4 | 14580 | 25.4 | 32570 | 25.0 | 3760 | 24.9 |
| Most deprived | 23540 | 41.0 | 62670 | 48.0 | 7650 | 50.7 |
| **History of hospital attendances[2]** | | | | | | |
| Adversity (violence, self-harm, substance misuse) | 2040 | 3.5 | 6555 | 5.0 | 685 | 4.5 |
| Mental health (exc. self-harm/substance misuse) | 905 | 1.6 | 3340 | 2.6 | 580 | 3.8 |
| Repeat A&E attendances (≥4) | 8125 | 14.1 | 21120 | 16.2 | 2790 | 18.5 |
| **Total linked to NPD[3]** | 48175 | 83.8 | 108420 | 83.1 | 12540 | 83.2 |
| Ever excluded, in pupil referral unit or alternative provision | 13615 | 28.3 | 32790 | 25.1 | 4020 | 32.1 |
| Ever recorded as persistently absent in a term | 19055 | 39.6 | 40560 | 31.1 | 4375 | 34.9 |
| Ever in care | 2330 | 4.8 | 6890 | 5.3 | 915 | 7.3 |
| Ever had child protection plan | 490 | 1.0 | 3890 | 3.0 | 940 | 7.5 |
| Achieved 5 A*-C GCSEs[4] | 8060 | 16.7 | 19940 | 15.3 | 2320 | 18.5 |
| **Total linked to NPD Census (FSM, SEN available)** | 47455 | 82.6 | 107420 | 82.3 | 12485 | 82.8 |
| Ever recorded as having Special Educational Needs | 20240 | 42.7 | 56500 | 52.6 | 7100 | 56.9 |
| Ever recorded as having Free School Meals | 20640 | 43.5 | 61360 | 57.1 | 8360 | 67.0 |

[1]Only including mothers aged ≤19 at last menstrual period;

[2]Hospital attendances within 2 years prior to 20 weeks of gestation;

[3]Social care and educational characteristics before 20 weeks of pregnancy;

[4]Including English and Maths, among mothers who were aged ≥16 at the start of the academic year in which they reached 20 weeks of pregnancy

## Discussion

This study assessed the impact of the FNP programme on all eligible mothers in England between 2010 and 2019, including the 25% who participated and the 75% who did not participate in the programme. We found no differences in child or maternal outcomes before, during or after the period in which the FNP was offered in the mothers' residential area. This finding suggests no overall harm and no detectable benefit on the outcomes measured during the time in which FNP was offered compared to other periods. Our findings add to the evidence base for the FNP in England, which demonstrate little evidence of an effect of the FNP on child

**Table 3. Relative risks comparing outcomes up to 2 years after delivery for eligible mothers during and after the FNP enrolment period, or living in an area in which FNP was never offered, relative to eligible mothers in sites before the FNP enrolment period.**

| | Total mothers | N with outcome | % with outcome | Adjusted Relative Risk* (95% CI) | | p-value |
|---|---|---|---|---|---|---|
| **Unplanned admissions for any diagnosis (child)** | | | | | | |
| Before FNP was active | 56375 | 20310 | 36.0 | reference | | |
| During FNP active period | 108045 | 39900 | 36.9 | 1.04 | (1.00, 1.08) | 0.06 |
| After FNP was active | 2415 | 895 | 37.1 | 1.04 | (0.80, 1.37) | 0.76 |
| **Unplanned admissions for maltreatment/injury (child)** | | | | | | |
| Before FNP was active | 56375 | 3020 | 5.4 | reference | | |
| During FNP active period | 108045 | 5755 | 5.3 | 0.99 | (0.92, 1.06) | 0.74 |
| After FNP was active | 2415 | 120 | 5.0 | 0.92 | (0.74, 1.14) | 0.45 |
| **Maternal admissions for adversity** | | | | | | |
| Before FNP was active | 57480 | 1050 | 1.8 | reference | | |
| During FNP active period | 110570 | 1895 | 1.7 | 0.91 | (0.79, 1.06) | 0.22 |
| After FNP was active | 2445 | 45 | 1.8 | 0.94 | (0.36, 2.50) | 0.91 |
| **Subsequent deliveries within 18 months of index birth** | | | | | | |
| Before FNP was active | 57480 | 4740 | 8.2 | reference | | |
| During FNP active period | 110570 | 9670 | 8.7 | 0.93 | (0.93, 1.06) | 0.82 |
| After FNP was active | 2445 | 245 | 10.0 | 0.65 | (0.95, 1.45) | 0.88 |

*Adjusted for maternal variables in Table 2

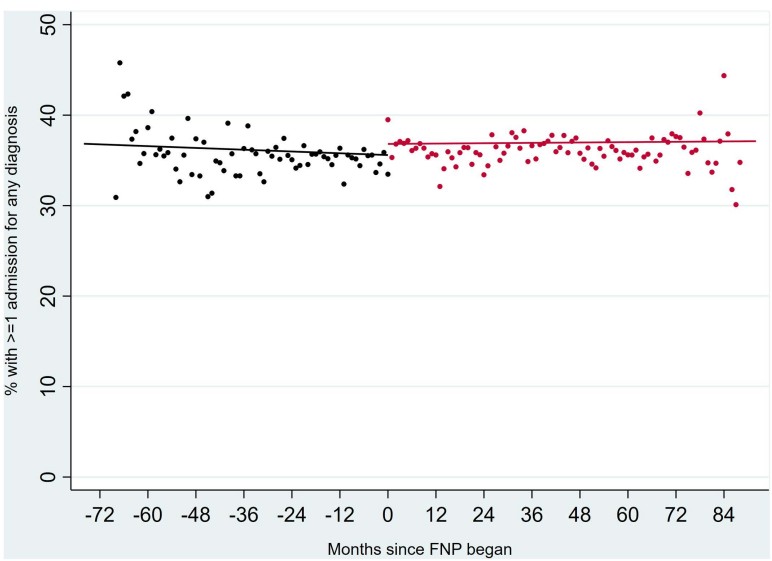

**Fig 1. Unplanned hospital admissions for any diagnosis in children up to two years after birth, comparing all eligible mothers before (black) and during (red) the period in which FNP was active.** Outcomes in the period after FNP had stopped are not plotted due to overlaps with the active FNP period across different LAs.

maltreatment outcomes in the medium term, but some evidence of improved school readiness associated with FNP participation.[10,16]

Strengths of our study include the large sample size representing the whole population of mothers aged 13-19 eligible for the FNP in England, and the use of linked information on health, education and children's social care. A first limitation is the possibility of time-varying

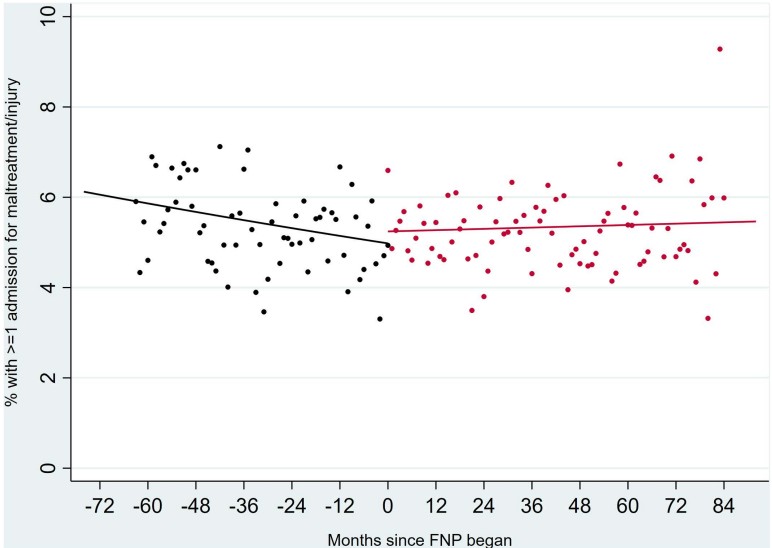

**Fig 2. Unplanned hospital admissions for maltreatment/ injury in children up to two years after birth, comparing all eligible mothers before (black) or during (red) the period in which FNP was active.** Outcomes in the period after FNP had stopped are not plotted due to overlaps with the active FNP period across different LAs.

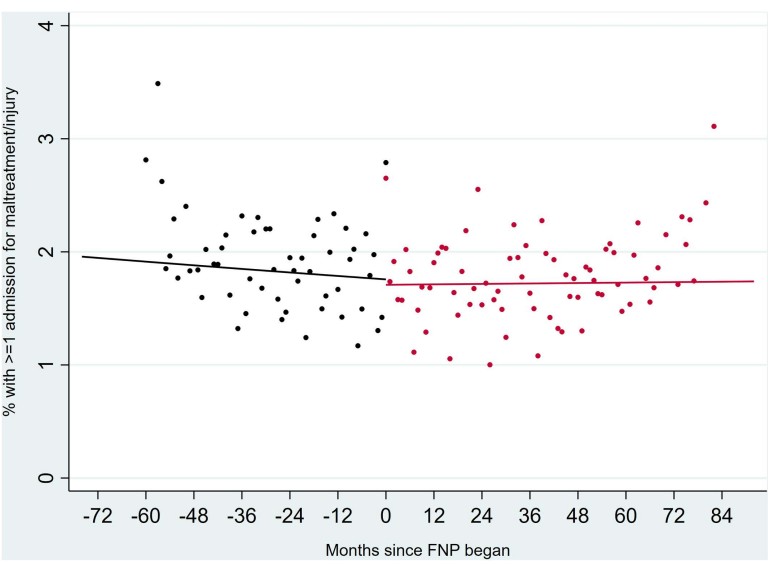

**Fig 3. Unplanned hospital admissions for adversity related diagnosis in mothers up to two years after birth, comparing all eligible mothers before (black) or during (red) the period in which FNP was active.** Outcomes in the period after FNP had stopped are not plotted due to overlaps with the active FNP period across different LAs.

confounding due to the before/during/after design. However, we accounted for calendar time and used interaction terms to identify any changes over time. We also evaluated outcomes for mothers in the period after enrolment had ended. A second limitation is that we were only able to control for the fairly crude maternal risk factors associated with enrolment in the FNP that were captured in the administrative data. We therefore cannot rule out unmeasured confounding, since we are comparing different populations of mothers across

time and geography. A third limitation is that we used relatively crude outcome measures of adversity-related unplanned hospital admissions, which may not capture benefits of FNP intervention for improved parenting. FNP practitioners report that mothers participating in the programme develop more reflective parenting and awareness of their child's needs: better measures of maternal well-being, confidence, mental health, parent–child interaction and child behaviour would allow us to understand more nuanced effects of the programme not routinely captured in administrative data. Finally, we were unable to account for the many other services that would have been available to adolescent mothers during each time period. [30] Further research could explore the extent to which are findings are replicable in other settings, e.g., in rural communities.

As rates of adolescent pregnancies have fallen over recent decades, adolescents who do become pregnant become a more selected population with a variety of needs (including, for example, appropriate social care involvement and support for mental health conditions) who are at risk of adverse outcomes for themselves and their children.[1,31] The FNP is one of the most promising interventions to support these mothers, but is only offered to a selected population of those who are eligible. We still do not know whether removing support for these mothers could lead to worse outcomes, taking into account increasing social disadvantage and the pressure on other services as a result of widespread health visitor shortages.[32] More needs to be done to understand which elements of intensive interventions are most effective, for whom and at which time points, and how to balance the use of highly intensive services available to small groups, with universal services that support all adolescent mothers.

Decisions about commissioning the Family Nurse Partnership need to weigh i) the strong local support for FNP, the benefits highlighted by qualitative research, and the lack of harms to the eligible population of adolescent mothers overall,[4,12,13] with ii) the lack of benefits observed for primary outcomes in the randomised controlled trials and observational evaluations of FNP, and the costs associated with offering this programme to a selected population of mothers (£1812 more than usual care per participant).[16,28,33,34] Most of the evidence of beneficial effects of the FNP comes from three USA trials of the Nurse Family Partnership, which showed mixed but overall positive impacts on child health and development outcomes and some maternal outcomes, along with a more recent trial in The Netherlands.[5–8,35] A more detailed synthesis of evidence can be found elsewhere.[27] In contrast, the Building Blocks trial in England showed no evidence of impact of FNP on most child outcomes, with the exception of some cognitive outcomes.[11,28] These differences are likely due in part to differences in usual care contexts between different countries (e.g., better overall access to services for adolescent mothers in England who are not enrolled in FNP. Another likely contributing factor is the differences in eligibility criteria for FNP in England compared with other countries. Young age is the main eligibility criteria for FNP in England, based on the ease of identifying the youngest adolescent mothers, associations between adolescent motherhood and social adversity, disrupted education and employment, and other factors contributing to poor birth and health outcomes among their children.[36–40] In contrast, the USA included additional socioeconomic criteria for enrolment, such as unemployment, low educational level or low income.[41–43] As a result, the population of young mothers enrolled in trials in other countries are a more selected and vulnerable group than in England, who may stand to benefit more from the FNP (as evidenced by greater effectiveness in socioeconomically deprived groups demonstrated in the USA trials).[8,44,45]

Our analysis provides a first step in considering the wider impact of the FNP, including on those not enrolled, when making decisions about commissioning this targeted intervention. Further research is also needed to understand whether other effects are present, where for

example the FNP might have an effect on the subsequent children born to mothers who were enrolled in the FNP with their first child.

## Supporting information

**S1 Table: Family Nurse Partnership site enrolment activity dates between April 2010 and March 2019 (mothers aged 13-19).**
(DOCX)

**S2 Table: ICD-10 code lists for maternal hospital admissions related to adversity, mental health, and chronic conditions.**
(DOCX)

**S3 Table: ICD-10 code lists for child maltreatment and health care utilisation-related outcomes.**
(DOCX)

**S4 Table: Maternal risk factors prior to 20 weeks of gestation used as covariates in adjusted models.**
(DOCX)

**S1 Fig: Flow diagram for participation in the study.**
(DOCX)

## Author contributions

**Conceptualization:** Katie Harron, Jan van der Meulen, Eilis Kennedy, Ruth Gilbert.

**Data curation:** Katie Harron, Francesca Cavallaro.

**Formal analysis:** Katie Harron.

**Funding acquisition:** Katie Harron.

**Investigation:** Katie Harron.

**Methodology:** Katie Harron.

**Supervision:** Katie Harron.

**Writing – original draft:** Katie Harron.

**Writing – review & editing:** Katie Harron, Francesca Cavallaro, Jan van der Meulen, Eilis Kennedy, Ruth Gilbert.

## Acknowledgements

We would like to thank members of our Study Steering Committee (Jane Barlow, Lorna Fraser, Emily Petherick, Marni Brownell, Loretta McGurry, and Romy Labrosse) for their helpful input, advice and suggestions throughout the course of the study.

The authors would also like to thank the FNP National Unit for their help in establishing the linked dataset and in interpretating the results, including Lynne Reed, Alisa Swarbrick, Andreea Calin, Sarah Tyndall, and Alex Stevenson. We would also like to thank the FNP nurses and clinical leads who provided valuable insight into the results and the discussion, including Cheryl Beirne, Alison Goodall, Amanda Malthouse, Nicole Hobson and Christine Anderson.

We would like to thank Sue Hillsden from the FNP National Unit for her help identifying catchment areas and activity dates for FNP sites.

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
