## [Decision Letter · Decision Letter 0]

15 Dec 2024

PONE-D-24-44718Effects of the Family Nurse Partnership on all eligible mothers: a data linkage cohort study in EnglandPLOS ONE

Dear Dr. Harron,

Thank you for submitting your manuscript to PLOS ONE. After careful consideration, we feel that it has merit but does not fully meet PLOS ONE’s publication criteria as it currently stands. Therefore, we invite you to submit a revised version of the manuscript that addresses the points raised during the review process.The manuscript has been evaluated by two reviewers, and their comments are available below.

The reviewers have raised a number of major concerns. They raise questions over the study design and request improvements to the reporting of methodological aspects of the study.

Could you please carefully revise the manuscript to address all comments raised?

Comments from PLOS Editorial Office: We note that one or more reviewers has referred to previously published works in their review. As always, we recommend that you please review and evaluate the requested works to determine whether they are relevant and should be cited. It is not a requirement to cite these works. We appreciate your attention to this request.

We look forward to receiving your revised manuscript.

Kind regards,

Helen Howard

Staff Editor

PLOS ONE

Journal Requirements:

This study was supported by funding from NIHR Health Services Delivery & Research (17/99/19), and the authors were in part supported by the NIHR through the Great Ormond Street Hospital Biomedical Research Centre and the NIHR Children and Families Policy Research Unit and Senior Investigator award for RG. 

This study was supported by funding from NIHR Health Services Delivery & Research (17/99/19), and

the authors were in part supported by the NIHR through the Great Ormond Street Hospital

Biomedical Research Centre and the NIHR Children and Families Policy Research Unit and Senior

Investigator award for RG.

We would like to thank members of our Study Steering Committee (Jane Barlow, Lorna Fraser, Emily

Petherick, Marni Brownell, Loretta McGurry, and Romy Labrosse) for their helpful input, advice and

suggestions throughout the course of the study.

The authors would also like to thank the FNP National Unit for their help in establishing the linked

dataset and in interpretating the results, including Lynne Reed, Alisa Swarbrick, Andreea Calin, Sarah

Tyndall, and Alex Stevenson. We would also like to thank the FNP nurses and clinical leads who

provided valuable insight into the results and the discussion, including Cheryl Beirne, Alison Goodall,

Amanda Malthouse, Nicole Hobson and Christine Anderson.

We would like to thank Sue Hillsden from the FNP National Unit for her help identifying catchment

areas and activity dates for FNP sites. 

 This study was supported by funding from NIHR Health Services Delivery & Research (17/99/19), and the authors were in part supported by the NIHR through the Great Ormond Street Hospital Biomedical Research Centre and the NIHR Children and Families Policy Research Unit and Senior Investigator award for RG. 

5. In the online submission form, you indicated that we are unable to share the individual data used for this study. HES and FNP data can be requested through NHS Digital and NPD can be requested through the Department for Education. 

Reviewers' comments:

Reviewer's Responses to Questions

**Comments to the Author**

1. Is the manuscript technically sound, and do the data support the conclusions?

Reviewer #1: Yes

Reviewer #2: No

2. Has the statistical analysis been performed appropriately and rigorously? 

Reviewer #1: Yes

Reviewer #2: No

3. Have the authors made all data underlying the findings in their manuscript fully available?

Reviewer #1: Yes

Reviewer #2: No

4. Is the manuscript presented in an intelligible fashion and written in standard English?

Reviewer #1: Yes

Reviewer #2: Yes

5. Review Comments to the Author

Reviewer #1: the manuscript is technically good, presented good introduction, method and data source. the result interpretation is soundly. the conclusion is soundly and as per the result. but the nature of the study might need further testing in different populations. for instance, rural communities.

Reviewer #2: Thank you for the opportunity to review this manuscript. While it addresses interesting issues, it fails to include fundamental information for readers to have in interpreting the results of this study:

1. FNP is a licensed program in which those licensed to deliver it are committed to protecting the licensed material. Thus, spill-over effects are not likely to be discerned, although local areas may choose to visit families in need more frequently than through usual health visiting and midwifery, and to use the general approach employed by FNP nurses during later periods of care, when FNP was not delivered.

2. It would be useful to examine the frequency of health-visiting and midwifery contacts before, during, and after FNP was delivered in specific areas. That would be useful data to use in this report, if the data are available. Evidence from the BB0-2 report indicated that health-visitors paid many more visits (~16) to the control group during the trial-period1 than the maximum typically delivered to families by health visitors prior to the trial (a maximum of 3). It’s not clear how many visits were paid to families with greater needs. Are health visitors and midwives currently visiting families in need today than before FNP went into effect?

3. The reader needs to see the degree to which those cases used in the final analyses were comparable on “baseline” characteristics used for comparing FNP enrollees to those adolescent mothers not enrolled, like the authors of this paper reported in a BMJ report published earlier this year. For a study like this one, these critical data points are needed by reviewers in the main body of the report to discern clearly which groups are being compared and the degree to which they were comparable.

4. The authors should acknowledge explicitly that FNP today focuses on those with heightened needs, given that the Cardiff trial results led to efforts on the part of FNP leaders to target the program on those with greater needs. Please search for the Adapt report.

5. Greater specification of the differences between those who received FNP and the rest of the population < 20Y will be important to report toward the beginning of this manuscript.

6. Note that a German trial of this program focused on those with overlapping needs but combined two versions of the program in the original reports: one delivered by midwives throughout the entire program period (pregnancy through child-age 2 – the Continuous model, and a second in which it was delivered by midwives from pregnancy through 6-months postpartum and then by social workers through child-age 2 - the tandem model. Early reports that combined these two models produced equivocal results.2 Evidence is now accumulating that the version of the program delivered by midwives continuously is beginning to show that the midwife-only model is superior to the tandem model. 3 The point is that the German replicators focused on those with overlapping needs predictive of child maltreatment. The same can be said about trials of this program conducted in the Netherlands4-6 and British Columbia,7 which again focused on those with overlapping needs, not simply maternal age <20. Note that the Netherlands and German trials were conducted in countries with extensive health and social care systems – like the UK.

7. It’s thus not surprising that FNP showed no evidence of beneficial carry-over effects, given the absence of detailed data on those served by FNP and the lack of comparability of those included in the comparison group. While the authors have made statistical adjustments using data available through data-linkage, the sources of such data are crude and do not account for unmeasured differences.

8. Moreover, questions should be raised about the choice of outcomes in this study: a) unplanned hospitalizations through child-age two, b) differences in child admissions for maltreatment/injury-related diagnoses and c) maternal admissions for adversity-related diagnoses. FNP nurses encourage parents to take their children to A&E to rule out internal injuries following an accident, and for mothers to do the same for themselves -- which will increase these types of admissions on the part of FNP-visited families. A better measure of child functioning (related to child maltreatment) is the number of days children were hospitalized with injuries, which was found to distinguish treatment and control groups in the US Memphis8 and the Canadian BC trials,7 which reflects injury severity, and which is less susceptible to FNP-visited families using GP and A&E care to protect their children’s health.

9. These issues should be addressed in the introduction of the manuscript and the discussion.

10. Moreover, the introduction of the manuscript should include the school-readiness finding from the BB2-6 report as well as the maternally reported language effects from BB0-2.

11. The rest of the manuscript depends upon the starting point – that is addressing the differences between those enrolled in the program and the rest of the population <20Y in comparing FNP enrollees to the rest of the populations within areas that delivered FNP.

12. Finally, the manuscript needs a careful editorial review for spelling and grammar.

References

1. Robling, M., Bekkers, M. J., Bell, K., Butler, C. C., Cannings-John, R., Channon, S., ... & Torgerson, D. (2016). Effectiveness of a nurse-led intensive home-visitation programme for first-time teenage mothers (Building Blocks): a pragmatic randomised controlled trial. The Lancet, 387(10014), 146- 155. 10.1016/S0140-6736(15)00392-X

2. Kliem, S., & Sandner, M. (2021). Prenatal and infancy home visiting in Germany: 7-year outcomes of a randomized trial. Pediatrics, 148(2). https://doi.org/10.1542/peds.2020-049610

3. Conti, G., Kliem, S., & Sandner, M. (2024). Early Home Visiting Delivery Model and Maternal and Child Mental Health at Primary School Age. AEA Papers and Proceedings, 114, 401–406.

4. Mejdoubi, J., van den Heijkant, S. C., van Leerdam, F. J., Heymans, M. W., Hirasing, R. A., & Crijnen, A. A. (2013). Effect of nurse home visits vs. usual care on reducing intimate partner violence in young high-risk pregnant women: a randomized controlled trial. PloS one, 8(10), e78185. https://doi.org/10.1371/journal.pone.0078185

5. Mejdoubi, J., van den Heijkant, S. C., van Leerdam, F. J., Crone, M., Crijnen, A., & HiraSing, R. A. (2014). Effects of nurse home visitation on cigarette smoking, pregnancy outcomes and breastfeeding: a randomized controlled trial. Midwifery, 30(6), 688-695. https://doi.org/10.1016/j.midw.2013.08.006

6. Mejdoubi, J., van den Heijkant, S. C., van Leerdam, F. J., Heymans, M. W., Crijnen, A., & Hirasing, R. A. (2015). The effect of VoorZorg, the Dutch nurse-family partnership, on child maltreatment and development: a randomized controlled trial. PLoS One, 10(4), e0120182. https://doi.org/10.1371/journal.pone.0120182

7. Catherine, N. L., MacMillan, H., Cullen, A., Zheng, Y., Xie, H., Boyle, M., ... & Waddell, C. (2024). Effectiveness of nurse‐home visiting in improving child and maternal outcomes prenatally to age two years: a randomised controlled trial (British Columbia Healthy Connections Project). Journal of Child Psychology and Psychiatry, 65(5), 644-655. https://doi.org/10.1111/jcpp.13846

8. Kitzman, H., Olds, D. L., Henderson, C. R., Hanks, C., Cole, R., Tatelbaum, R., ... & Barnard, K. (1997). Effect of prenatal and infancy home visitation by nurses on pregnancy outcomes, childhood injuries, and repeated childbearing: a randomized controlled trial. JAMA, 278(8), 644-652. doi:10.1001/JAMA.1997.03550080054039

6. PLOS authors have the option to publish the peer review history of their article (what does this mean? ). If published, this will include your full peer review and any attached files.

**Do you want your identity to be public for this peer review?** For information about this choice, including consent withdrawal, please see our Privacy Policy .

Reviewer #1: No

Reviewer #2: No

---

## [Author Response · Author response to Decision Letter 1]

17 Jan 2025

Please see uploaded response to reviewers.

---

## [Decision Letter · Decision Letter 1]

12 Feb 2025

PONE-D-24-44718R1Effects of the Family Nurse Partnership on all eligible mothers: a data linkage cohort study in EnglandPLOS ONE

Dear Dr. Harron,

Thank you for submitting your manuscript to PLOS ONE. After careful consideration, we feel that it has merit but does not fully meet PLOS ONE’s publication criteria as it currently stands. Therefore, we invite you to submit a revised version of the manuscript that addresses the points raised during the review process.

We look forward to receiving your revised manuscript.

Kind regards,

Mu-Hong Chen, M.D., Ph.D.

Academic Editor

PLOS ONE

Journal Requirements:

Reviewers' comments:

Reviewer's Responses to Questions

**Comments to the Author**

1. If the authors have adequately addressed your comments raised in a previous round of review and you feel that this manuscript is now acceptable for publication, you may indicate that here to bypass the “Comments to the Author” section, enter your conflict of interest statement in the “Confidential to Editor” section, and submit your "Accept" recommendation.

Reviewer #3: (No Response)

2. Is the manuscript technically sound, and do the data support the conclusions?

Reviewer #3: Yes

3. Has the statistical analysis been performed appropriately and rigorously? 

Reviewer #3: Yes

4. Have the authors made all data underlying the findings in their manuscript fully available?

Reviewer #3: Yes

5. Is the manuscript presented in an intelligible fashion and written in standard English?

Reviewer #3: Yes

6. Review Comments to the Author

Reviewer #3: Overall, the article presents its methodology and study limitations clearly. This study reports rather unoptimistic findings: there is no evidence of differences in unplanned hospital admissions between children born during the Family Nurse Partnership (FNP) period, those born after FNP was implemented, and those born before FNP started. Although I initially found it unclear why unplanned hospital admissions were chosen as the primary outcome, the authors explained in the methods section that this decision was based on findings from a previous study, which makes sense.

To be honest, I am impressed by the authors' courage in presenting such disappointing data regarding the FNP program. Therefore, I believe the discussion section could be more in-depth or inspiring, which might enhance the readability of the article. For example, the second paragraph of the introduction mentions that the FNP program implemented in the United States was effective, making one curious about the differences in implementation between the UK and the US that might have led to the variation in outcomes. Additionally, I am curious about incorporating cost-effectiveness data, which might help readers assess the feasibility of the FNP program.

7. PLOS authors have the option to publish the peer review history of their article (what does this mean? ). If published, this will include your full peer review and any attached files.

**Do you want your identity to be public for this peer review?** For information about this choice, including consent withdrawal, please see our Privacy Policy .

Reviewer #3: No

---

## [Author Response · Author response to Decision Letter 2]

19 Feb 2025

Dear Editors

Thank you for the additional reviews on our manuscript. The final reviewer had two main points, which we have responded to below.

Comment 1: Overall, the article presents its methodology and study limitations clearly. This study reports rather unoptimistic findings: there is no evidence of differences in unplanned hospital admissions between children born during the Family Nurse Partnership (FNP) period, those born after FNP was implemented, and those born before FNP started. Although I initially found it unclear why unplanned hospital admissions were chosen as the primary outcome, the authors explained in the methods section that this decision was based on findings from a previous study, which makes sense.

Response: Thank you for this positive feedback.

To be honest, I am impressed by the authors' courage in presenting such disappointing data regarding the FNP program. Therefore, I believe the discussion section could be more in-depth or inspiring, which might enhance the readability of the article. For example, the second paragraph of the introduction mentions that the FNP program implemented in the United States was effective, making one curious about the differences in implementation between the UK and the US that might have led to the variation in outcomes. Additionally, I am curious about incorporating cost-effectiveness data, which might help readers assess the feasibility of the FNP program.

Response: We agree that we could give more information in the discussion section about why there are differences in the evidence on the effectiveness of the FNP in the US and the UK, and have now amended the text. Although looking at cost-effectiveness is beyond the scope of this study, we also now refer to a cost-effectiveness study of the FNP in England, for context.

We hope that these changes satisfy the reviewer and the editors, but welcome any further feedback.

Kind regards,

Katie Harron

---

## [Decision Letter · Decision Letter 2]

25 Feb 2025

Effects of the Family Nurse Partnership on all eligible mothers: a data linkage cohort study in England

PONE-D-24-44718R2

Dear Dr. Katie Harron,

We’re pleased to inform you that your manuscript has been judged scientifically suitable for publication and will be formally accepted for publication once it meets all outstanding technical requirements.

Kind regards,

Mu-Hong Chen, M.D., Ph.D.

Academic Editor

PLOS ONE

Additional Editor Comments (optional):

Reviewers' comments:

Reviewer's Responses to Questions

**Comments to the Author**

1. If the authors have adequately addressed your comments raised in a previous round of review and you feel that this manuscript is now acceptable for publication, you may indicate that here to bypass the “Comments to the Author” section, enter your conflict of interest statement in the “Confidential to Editor” section, and submit your "Accept" recommendation.

Reviewer #3: All comments have been addressed

2. Is the manuscript technically sound, and do the data support the conclusions?

Reviewer #3: Yes

3. Has the statistical analysis been performed appropriately and rigorously? 

Reviewer #3: Yes

4. Have the authors made all data underlying the findings in their manuscript fully available?

Reviewer #3: Yes

5. Is the manuscript presented in an intelligible fashion and written in standard English?

Reviewer #3: Yes

6. Review Comments to the Author

Reviewer #3: I appreciate the careful changes to your latest manuscript. Several major modifications make the new discussion section more sophisticated and applicable. In particular, international comparisons help explain why FNP affects vary among countries. In addition, cost-effectiveness considerations enrich the manuscript by providing context for assessing FNP implementation viability. These improvements make the discussion more thorough and enlightening, especially for policymakers considering FNP implementation or modification. Thank you for your attention to these details—I think they improve the manuscript.

7. PLOS authors have the option to publish the peer review history of their article (what does this mean? ). If published, this will include your full peer review and any attached files.

**Do you want your identity to be public for this peer review?** For information about this choice, including consent withdrawal, please see our Privacy Policy .

Reviewer #3: No

---

## [Editor Report · Acceptance letter]

PONE-D-24-44718R2

PLOS ONE

Dear Dr. Harron,

I'm pleased to inform you that your manuscript has been deemed suitable for publication in PLOS ONE. Congratulations! Your manuscript is now being handed over to our production team.

Kind regards,

on behalf of

Dr. Mu-Hong Chen

Academic Editor

PLOS ONE